# CuMV VLPs Containing the RBM from SARS-CoV-2 Spike Protein Drive Dendritic Cell Activation and Th1 Polarization

**DOI:** 10.3390/pharmaceutics15030825

**Published:** 2023-03-02

**Authors:** Ana Isabel Sebastião, Daniela Mateus, Mylène A. Carrascal, Cátia Sousa, Luísa Cortes, Martin F. Bachmann, Anália do Carmo, Ana Miguel Matos, Maria Goreti F. Sales, Maria Teresa Cruz

**Affiliations:** 1Faculty of Pharmacy, University of Coimbra, 3000-548 Coimbra, Portugal; 2Center for Innovative Biomedicine and Biotechnology (CIBB), University of Coimbra, 3000-548 Coimbra, Portugal; 3Tecnimede Group, 2710-089 Sintra, Portugal; 4Department of BioMedical Research, University of Bern, 3012 Bern, Switzerland; 5Clinical Pathology Department, Centro Hospitalar e Universitário de Coimbra, 3000-075 Coimbra, Portugal; 6Chemical Engineering Processes and Forest Products Research Center, CIEPQPF, Faculty of Sciences and Technology, University of Coimbra, 3030-790 Coimbra, Portugal; 7BioMark@UC/CEB—LABBELS, Department of Chemical Engineering, Faculty of Sciences and Technology, University of Coimbra, 3030-790 Coimbra, Portugal

**Keywords:** dendritic cells, SARS-CoV-2, COVID-19, memory T cell, adaptive immune response, virus-like particles

## Abstract

Dendritic cells (DCs) are the most specialized and proficient antigen-presenting cells. They bridge innate and adaptive immunity and display a powerful capacity to prime antigen-specific T cells. The interaction of DCs with the receptor-binding domain of the spike (S) protein from the severe acute respiratory syndrome coronavirus 2 (SARS-CoV-2) is a pivotal step to induce effective immunity against the S protein-based vaccination protocols, as well as the SARS-CoV-2 virus. Herein, we describe the cellular and molecular events triggered by virus-like particles (VLPs) containing the receptor-binding motif from the SARS-CoV-2 spike protein in human monocyte-derived dendritic cells, or, as controls, in the presence of the Toll-like receptors (TLR)3 and TLR7/8 agonists, comprehending the events of dendritic cell maturation and their crosstalk with T cells. The results demonstrated that VLPs boosted the expression of major histocompatibility complex molecules and co-stimulatory receptors of DCs, indicating their maturation. Furthermore, DCs’ interaction with VLPs promoted the activation of the NF-kB pathway, a very important intracellular signalling pathway responsible for triggering the expression and secretion of proinflammatory cytokines. Additionally, co-culture of DCs with T cells triggered CD4^+^ (mainly CD4^+^Tbet^+^) and CD8^+^ T cell proliferation. Our results suggested that VLPs increase cellular immunity, involving DC maturation and T cell polarization towards a type 1 T cells profile. By providing deeper insight into the mechanisms of activation and regulation of the immune system by DCs, these findings will enable the design of effective vaccines against SARS-CoV-2.

## 1. Introduction

One of the most disastrous pandemics in human history was caused by the severe acute respiratory syndrome coronavirus 2 (SARS-CoV-2), a virus that led to the coronavirus disease 19 (COVID-19) [1]. Similar to other coronaviruses, SARS-CoV-2 is an enveloped and positive-stranded RNA virus ((+)ss-RNA), and the biggest of all viruses of the beta-coronavirus subfamily (27 to 32 kbp). The SARS-CoV-2 (+)ss-RNA has six open reading frames (ORFs), at least, and each one codifies a specific protein, one of them being the spike (S) protein [2,3,4]. The S protein is composed of two main functional domains, S1 and S2, whose functions are recognizing and binding to specific cell surface receptors and inducing envelope fusion with the target cell membrane [3]. The S1 domain comprises a receptor-binding domain (RBD), which recognizes and links to its target receptor, specifically the angiotensin-converting enzyme 2 (ACE-2) [5]. The host ACE-2 and the transmembrane serine protease TMPRSS2 allow the perception and the priming of SARS-CoV-2 S protein, respectively [6]. At the end of the replication cycle, SARS-CoV-2 induces cellular pyroptosis and triggers the release of both danger-associated molecular patterns (DAMPs) and pathogen-associated molecular patterns (PAMPs). DAMPs and PAMPs bind to specific receptors, for example Toll-like receptors (TLRs), displayed in cells from the innate immune system, which leads to proinflammatory molecules’ release, which may result in a cytokine storm [3,7]. Thus, a thorough understanding of the cellular and humoral mechanisms underlying antiviral immunity against SARS-CoV-2 is crucial for designing vaccination strategies.

Dendritic cells (DCs) are the most potent antigen-presenting cells (APCs) in the innate immune system, with the ability to communicate with the adaptive immune system after the recognition and response to both DAMPs and PAMPs [8]. Antigen recognition triggers a maturation process, which induces functional changes. Mature dendritic cells (mDCs) arrive at lymph nodes and present the processed antigens to naïve T cells [9,10]. DCs express various pattern recognition receptor families, including TLRs and cytosolic retinoic acid-inducible gene (RIG)-I-like receptors, which are activated upon infection or interaction with viruses [11]. DCs can detect infection and guide T and B cells for efficient anti-SARS-CoV-2 immune responses.

Even though the activity of several immune cells in the infection by SARS-CoV-2 is already known, such as macrophages and monocytes [12], the role of DCs is not fully characterized. It is known that RBD (and S protein, but to less extent) activates DCs, promoting their maturation [13]. However, a more complete characterization of this maturation status and which signalling pathways and immune responses are triggered still needs to be elucidated. In order to fill this gap, the purpose of this work consists of describing the pathways DCs undergo during the interaction with virus-like particles (VLPs) containing the receptor-binding motif (RBM) from the SARS-CoV-2 S protein, comprehending the events from initial DC maturation until their crosstalk with T cells. Indeed, in previous studies, different VLPs showed an ability to modulate mice DCs-mediated immune response [14,15]. Hence, this study allows for deeper insights into the activation mechanisms of the immune system, as well as its regulation by human DCs, providing valuable tools for counteracting SARS-CoV-2 and for designing effective vaccines.

## 2. Materials and Methods

### 2.1. Dendritic Cells Obtention and Culture

DCs were obtained and cultured according to methods already described by our laboratory [16]. Peripheral blood mononuclear cells (PBMCs) were isolated through Ficoll-Paque (GE Healthcare, Chalfont St. Giles, UK) gradient centrifugation from buffy coats of healthy volunteers. Monocytes were isolated by positive selection using CD14 antibody-coated magnetic beads (Miltenyi Biotec, Bergisch Gladbach, Germany and BioLegend, London, UK) and T cells using CD3 antibody-coated magnetic beads (Miltenyi Biotec), as described by the manufacturer. T cells were frozen at −80 °C using a solution of foetal bovine serum (FBS), 5% of glucose at 40%, and 10% of dimethyl sulphoxide (DMSO) until co-culture with DCs. Monocytes were cultured in RPMI 1640 (Gibco, Waltham, MA, USA) supplemented with 10% heat-inactivated FBS, 100 U/mL penicillin, 1 mM sodium pyruvate and MEM non-essential amino acids, 100 µg/mL streptomycin, and 2 mM glutamax, (all from Gibco). Then, 1 × 10^6^ monocytes/mL were differentiated into immature DCs (iDCs) in culture media supplemented with 400 U/mL of granulocyte-macrophage colony-stimulating factor (GM-CSF) (Peprotech, London, UK) and 250 U/mL of IL-4 (Peprotech), which was refreshed every 2 days. DC maturation was induced on the 6th day of culture by the addition of 10 μg/mL of virus-like particles (VLPs) (kindly provided by Martin F. Bachmann [17]), in the presence or in the absence of 20 μg/mL of polyinosinic:polycytidylic acid (Poly I:C), a TLR3 agonist (Novus Biologicals, Abingdon, UK), and 2.5 μg/mL of Resiquimod (R848), a TLR7/8 agonist (Sigma-Aldrich, St. Louis, MO, USA).

### 2.2. VLPs Generation

The VLPs used were derived from cucumber mosaic virus (CuMV9 and contained naturally packaged RNA derived from *Escherichia coli*, which was incorporated during bacterial expression [18], and the RBM within RBD was also genetically incorporated [17]. VLP generation was according to procedures already described [17]. Briefly, *E. coli* C2566 (New England Biolabs, Ipswich, MA, USA) cells were transformed with the pETDu-CMVB3d-nCoV-M-CMVTT plasmid. The clones with the highest expression level of target proteins (mosaic CuMVTT-RBM) were cultured in 2TY medium containing ampicillin (100 mg/L) and induced with 0.2 mM IPTG. The resulting biomass was collected and frozen at −20 °C. To disrupt the cells, the biomass was resuspended in a pH 8.0 buffer, which contained EDTA, Et-SH, glycerol, Tris, and sucrose, then treated with ultrasound on ice. Later, 0.5% Triton X-100 was added and rotated without centrifugation. This biomass was then clarified for 10 min, at 10,000 rpm, and with disposal of the pellet. Next, the soluble fraction was submitted to a sucrose gradient centrifugation. The gradient fractions were removed and the CuMV VLP-containing fraction was diluted 1:1 with 2 mM EDTA, 20 mM Tris, 5% glycerol (pH 8.0). The VLPs were sedimented, and the pellet was dissolved. Endotoxin measurement was performed, and the produced vaccine showed ~50 EU/mg. Mosaic CuMVTT-RBM vaccine candidate was next characterized using agarose gel, dynamic light scattering, electron microscopy, and sodium dodecyl sulphate-polyacrylamide gel electrophoresis (SDS-PAGE). Protein concentration was determined using the bicinchoninic acid assay test.

### 2.3. Dendritic Cells Maturation

Fluorescence-conjugated antibodies were used to stain DCs: ACE2-APC, CD1a-Alexa Fluor 488, CD40-APC, CD80-PerCP/Cy5.5, CD83-PE, CD86-Alexa Fluor 488, C-X-C chemokine receptor type 4 (CxCR4)-PE, dendritic cell-specific intercellular adhesion molecule-3-grabbing non-integrin (DC-SIGN)/CD209-PerCP/Cy5.5, human leukocyte antigen (HLA)-DR-PE, HLA-ABC-APC, and programmed death-ligand 1 (PD-L1)-FITC (all from BioLegend). Isotype-matched antibodies were used as controls. Shortly, DCs were washed and resuspended in phosphate-buffered saline (PBS) + 1% FBS. Then, 3 µL of fluorescence-conjugated antibodies were added to cells and incubated for 30 min, at 4 °C, in the dark. Cells were subsequently washed, resuspended in PBS + 1% FBS, and analysed in an Accuri C6 flow cytometer (BD Bioscience, San Jose, CA, USA). Data were analysed with GraphPad Prism version 8 (GraphPad Software, San Diego, CA, USA), and the results are presented as mean fluorescence intensity (MFI), obtained by the subtraction of isotype control values.

### 2.4. Cytokine Production

The transcriptional levels of interleukin (*IL*)*6*, *IL10*, *IL15*, *IL18*, *IL12a*, *IL12b*, *IL1β*, C-C chemokine receptor 7 (*CCR7*)*,* C-C motif ligand 22 (*CCL22*), transforming growth factor-beta (*TGFβ*)*,* tumour necrosis factor (*TNF*)*α,* and inducible nitric oxide synthase (*iNOS*) were analysed using quantitative real-time polymerase chain reaction (qRT PCR) on maturated DCs (Appendix A: Primer sequences for studied genes). Ribonucleic acid (RNA) was extracted with an NZY Total RNA Isolation kit (Nzytech, Lisbon, Portugal), according to the manufacturer’s instructions. Then, using a NanoDrop spectrophotometer (ThermoScientific, Wilmington, DE, USA), RNA concentration was measured using OD260. The samples were kept in RNA Storage Solution (Ambion, Foster City, CA, USA) at −80 °C until further use. Complementary deoxyribonucleic acid (cDNA) was obtained by reverse transcription of 1 µg of total RNA using the NZY First-Strand cDNA Synthesis Kit (Nzytech). qPCR reactions were performed on Bio-Rad CFX Connect equipment (Biorad, Hercules, CA, USA). The built-in CFX Maestro software (Bio-Rad Laboratories, Hercules, CA, USA) was used to analyse gene transcription changes. The results were normalized using the TATA-box binding protein (TBP) as a reference gene, once this gene was experimentally determined as the most stable for the used treatment conditions after using Genex software (MultiD Analyses AB, Göteberg, Sweden).

The secretion of IL-1β, IL-10, and IL-12p70 by mature DCs and interferon (IFN)-γ and IL-4 by T cells after co-culture with matured DCs was analysed using Enzyme-Linked Immunosorbent Assay (ELISA) Max Deluxe Kits (Biolegend, London, UK), according to the manufacturer’s instructions.

### 2.5. Immunofluorescence and Confocal Microscopy

A µ-Slide 8-well ibidi chamber (Ibidi GmbH, Gräfelfing, Germany) was covered with 300 μL of L-Polilisine (Sigma-Aldrich) and left overnight at room temperature. After that, L-Polilisine was removed, and 0.3 × 10^6^ DCs/mL was seeded in each well and stimulated at different time points: 30, 45, 60, and 120 min by adding 10 μg/mL of VLPs, 20 μg/mL of Poly I:C (Novus Biologicals, Abingdon, UK), and 2.5 μg/mL of R848 (Sigma-Aldrich). Then, wells were washed once with PBS, and DCs were fixed with 4% paraformaldehyde for 15 min at room temperature and washed three times with PBS 1% + glycine 0.1 M. Cells were then blocked for 1 h with 5% goat serum + 0.3% Triton X-100 in PBS and further incubated with the primary rabbit monoclonal anti-nuclear factor kappa-light-chain-enhancer of activated B cells (NF-kB) p65 XP^®^ antibody, at the dilution of 1:400 (Cell Signaling Technology, Inc., Danvers, MA, USA), overnight at 4 °C. After washing three times with PBS, cells were incubated for 1 h at room temperature with the anti-rabbit horseradish peroxidase (HRP)-conjugated secondary antibodies, diluted at 1:400 (Santa Cruz Biotechnology, Dallas, TX, USA). Then, cells were washed three times with PBS and stained with Hoechst dye 0.5 µg/mL (Invitrogen, Thermo Fisher Scientific, Wilmington, DE, USA), for 5 min at 37 °C, to stain the nucleus. Controls were stained with only secondary antibodies. Stained cells were washed with PBS once, mounted with mounting medium (Ibidi GmbH, Gräfelfing, Germany), and further observed in a Zeiss LSM710 confocal microscope (Carl Zeiss Microscopy GmbH, Jena, Germany) equipped with a Plan-Apochromat 40×/1.4 objective, and using the 488 nm (Argon/2 laser) and 405 nm (Diode 405-30) laser lines to image Alexa-488 (NFKB) and DAPI, respectively. Images were analysed using open source software FIJI [19].

### 2.6. Early Signaling Assay and Western Blot Analysis

Cells lysis was performed by incubating cells for 30 min on ice with a phosphatase inhibitor cocktail (PhosSTOP, Roche Diagnostics, Mannheim, Germany) and ice-cold radioimmunoprecipitation assay (RIPA) buffer (protease inhibitor cocktail (Complete, Mini, Roche Diagnostics, Mannheim, Germany)), 150 mM sodium chloride, 50 mM Tris (pH 7.5), 5 mM ethylene glycol-bis (2-aminoethylether)-N,N,N0, N0 -tetraacetic acid, 0.5% sodium deoxycholate, 0.1% sodium dodecyl sulphate (SDS), and 1% Triton X-100. The lysates were centrifuged for 10 min at 14,000 rpm at 4 °C. The supernatants were collected and stored at −20 °C until further use. Protein concentration was determined with the bicinchoninic acid kit (Sigma-Aldrich Co.) and 30 μg of total cell protein was denatured at 95 °C for 5 min in a buffer (composed of bromophenol blue, 20% glycerol, 10% 2-mercaptoethanol, 5% SDS and 0.125 M Tris–HCl pH 6.8). Proteins were separated by SDS-PAGE and electrotransferred using a wet transfer system at 350 mA for 210 min onto polyvinylidene difluoride (PVDF) membranes. Then, the PVDF membranes were blocked with 5% non-fat dry milk in Tris-buffered saline (TBS)-Tween 20 (0.1%) for 1 h and probed overnight at 4 °C with rabbit polyclonal antibodies, diluted at 1:1000, against the inhibitor of NF-kB (IkBα) (Cell Signaling Technology, Inc., Danvers, MA, USA). The membranes were washed with TBS-Tween 20 and incubated with anti-rabbit HRP-conjugated secondary antibodies for 1 h at room temperature, using a dilution of at 1:5000 (Santa Cruz Biotechnology, Dallas, TX, USA). After washing, immune complexes were detected with enhanced chemiluminescence reagent in the imaging system ImageQuant LAS 500 (GE Healthcare). The loading control β-Tubulin I was also detected after reprobing the membranes for 1 h at room temperature with a mouse monoclonal anti-β-Tubulin I antibody (Sigma-Aldrich Co.), diluted at 1:20,000. Image analysis was performed with Total Lab TL120 (Nonlinear Dynamics Ltd., Edmonton, AB, Canada). Densitometric values of protein were divided by that of the loading control Tubulin to determine loading amounts. Normalized values were determined by considering the basal condition = 1.

### 2.7. Mixed Lymphocyte Reaction (MLR)

Autologous T cells were co-cultured with matured DCs for 5 days at a 10:1 ratio, after being submitted to a staining protocol with carboxyfluorescein succinimidyl ester (CFSE). All co-cultures were carried out in U-bottomed 96-well plates in a final volume of 200 µL of RPMI medium. At the end of the co-culture period, cells were stained with fluorescence-conjugated antibodies, namely CD4-PerCP/Cy5.5 and CD8-APC antibodies (BioLegend) in order to evaluate the percentage of positive T cell proliferation. Type 1 T cells (Th1), type 2 T cells (Th2), and regulatory T cells (Treg) subsets were also evaluated through flow cytometry after the co-culture period with DCs for 5 days. The autologous T cells were stained using anti-CD4-PerCP/Cy5.5, anti-CD8-APC, anti-CD25-APC, anti-forkhead-box-P3 (FoxP3)-FITC, anti-GATA-binding protein 3 (GATA3)-FITC, and anti-T-box protein expressed in T cells (T-bet)-PE (Biolegend). As some markers are intracellular, the Cyto-Fast™ Fix/Perm Buffer Set (BioLegend), a fixation and cell permeabilization kit, was used for the intracellular staining, according to the manufacturer’s instructions. Data were analysed with GraphPad Prism version 8 (GraphPad Software, San Diego, CA, USA) and the results are presented as a percentage of positive cells (%) after subtraction of isotype control values.

### 2.8. Statistical Analysis

Statistical analysis was performed using GraphPad Prism, version 8 (GraphPad Software, San Diego, CA, USA). Data are shown as mean ± standard error of the mean (SEM) of the indicated number of experiments. Comparisons were made using multiple group comparisons by one-way ANOVA analysis, with a Tukey multiple comparison post-test (a Dunnett comparison post-test in Western blot results). Significance levels are as follows: * *p* < 0.05, ** *p* < 0.01, *** *p* < 0.001, **** *p* < 0.0001.

## 3. Results

### 3.1. VLPs Containing the RBM from SARS-CoV-2 Spike Protein Promote DCs Maturation and Activation

Upon antigen recognition by iDCs, they initiate a maturation process that includes cytokine secretion and surface molecule expression, leading to the development of migratory and co-stimulatory abilities. As a result, DCs develop typical characteristics of APCs. To address how human DCs react to VLPs containing the RBM from SARS-CoV-2 S protein, we promoted the differentiation of primary DCs from monocytes of healthy donors and challenged them with VLPs packaged with natural, single-stranded *E. coli*-derived RNA [18]. Treatment with Poly I:C and R848 was included as a control in order to assess the state and responsivity of the iDCs, and to disclose whether they potentiate the response triggered by VLPs. Therefore, DCs were analysed for their expression levels of antigen recognition molecules (DC-SIGN, ACE2, and CD1a), co-stimulatory (CD40, CD80, CD83, and CD86), and co-inhibitory molecules (PD-L1), as well as chemotactic receptors (CxCR4) [20,21,22,23].

Overall, the results demonstrated that VLPs induce a statistically significant increase in the protein levels of the co-stimulatory molecules CD40, CD83, and CD86 (Figure 1A,C,D). However, when stimulated with VLPs plus Poly I:C and R848, DCs present the most extensive maturation status. This increase in the detected MFI is statistically significant relative to the CTR in all the assessed co-stimulatory molecules. Accompanying the increase in co-stimulatory molecules, the protein levels of antigen-presenting molecules MHC-I and MHC-II are also heightened, especially MHC-II, such that all stimuli lead to a statistically significant increase in MHC-II relative to the CTR (Figure 1F). PD-L1 is an immune checkpoint, so its upregulation will help to keep the balance in the immune response [24]. Interestingly, VLPs induced a statistically significant increase in PD-L1 protein levels when compared to the CTR (Figure 1G), while the remaining conditions did not achieve this effect.

The chemokine receptor CxCR4 is related to DC migration towards the lymph nodes [22,25], and only VLPs induced a statistically significant increase in CxCR4 relatively to CTR (Figure 1H).

In contrast to all the upregulations that have been described so far, DC-SIGN and ACE2 expression levels were reduced. In particular, the expression of ACE2 was reduced in all conditions with statistical significance when compared to CTR (Figure 1L). In turn, DC-SIGN expression was also reduced with statistical significance compared to the CTR after DC stimulation with VLPs and VLPs plus Poly I:C and R848 (Figure 1J).

Finally, the reduction in the CD1a expression is often related to the reduction in the immature state of DCs [26]. Indeed, the results obtained suggest a tendency to have its expression downregulated after DC stimulation (Figure 1K). Explicitly, the condition VLPs plus Poly I:C and R848 presented a statistically significant decrease in the CD1a levels compared to the CTR. This means that there is a tendency for CD1a levels to be reduced after DC stimulation. Taken together, these results indicate that DCs acquired a mature profile in the presence of VLPs and Poly I:C and R848.

### 3.2. VLPs Containing the RBM from SARS-CoV-2 Spike Protein with Poly I:C and R848 Tend to Stimulate a Proinflammatory Activation Profile in DCs

When DCs are activated and matured in response to antigenic recognition, they produce specific cytokines that are dependent on the nature of the pathogen. Cytokine production by DCs is crucial for naïve T cell activation and polarization. To investigate DCs’ activation profile induced by VLPs, we evaluated a set of proinflammatory cytokines and mediators at the mRNA level, namely *CCL22*, *CCR7*, *IL1β*, *IL6*, *IL12a*, *IL12b*, *IL15*, *IL18*, *iNOS*, and *TNFα*, as well as anti-inflammatory mediators, specifically *IL10* and *TGFβ*.

VLPs plus Poly I:C and R848 induced a statistically significant increase in the gene transcription of *CCL22* when compared to CTR (Figure 2a). As there is a switch in the chemokine receptor profile in mature DCs, which allows the relocation of those cells from inflamed tissues towards draining lymph nodes [27], the expression of *CCR7* was evaluated. In fact, the transcriptional levels of *CCR7* tend to increase significantly upon VLPs plus Poly I:C and R848 stimuli (Figure 2B).

In response to VLPs plus Poly I:C and R848 stimuli, the transcriptional levels of the proinflammatory genes *IL12B*, *IL15* and *IL18* showed a statistically significant increase compared to CTR (Figure 2F–H). Furthermore, the anti-inflammatory genes *IL10* and *TGFβ* (Figure 2K–L) tend to show decreased transcriptional levels, without statistical significance.

Subsequently, we examined whether the increase in the transcription of *IL-1β* and *IL-12p70* was accompanied by an increase in the release of these cytokines. The release of IL-10 was also evaluated. Of note, IL-12 and IL-10 contribute to the efficient polarization of naïve T cells towards the Th1 and Treg phenotypes, respectively [28,29]. A statistically significant increase in the release of all evaluated cytokines was induced by VLP plus Poly I:C and R848 (Figure 3A–C). Despite the fact that Poly I:C and R848 induced a similar profile, only proinflammatory cytokines had increased release with statistical significance (Figure 3A,B).

Taken together, these data suggest that VLPs containing the RBM from SARS-CoV-2 S protein trigger a proinflammatory profile on DCs.

### 3.3. VLPs Containing the RBM from SARS-CoV-2 Spike Protein Trigger NF-kB Signalling Pathway in DCs

To reach a deep understanding of the mechanisms underlying DCs’ activation and maturation, we explore whether the NF-kB signalling pathway is triggered by VLPs containing the RBM from SARS-CoV-2 S protein. Lipopolysaccharide (LPS) was employed to validate the functional status of iDCs as a positive control. The canonical NF-kB pathway activation comprises the recognition of a molecule by a cell receptor (such as LPS being recognized by TLR4), which induces the activation of the IkB kinase complex. This activation, in turn, leads to the phosphorylation of the IκBα and its degradation, resulting in the release of the NF-kB dimers, the p50 and p65 proteins. This release leads to the translocation of NF-kB dimmers to the nucleus, culminating in several transcriptional modifications in target genes [30]. It is widely known that the described pathway promotes the release of proinflammatory cytokines and the DC maturation process, which includes the upregulation of the expression of MHC-II and co-stimulatory molecules [31].

Since the degradation of IkBα is crucial for the NF-kB activation, we first assessed whether the previous stimuli triggered IkBα degradation. After 30 min, LPS (1 μg/mL) and Poly I:C and R848 had almost completely degraded IkBα. VLPs seem to induce partial and transitory IkBα phosphorylation that peaked at 120 min (Figure 4A). Additionally, the nuclear translocation of the p65 subunit was confirmed by immunocytochemistry. As expected, the CTR exhibited all the p65 immunoreactivity concentrated in the cytoplasm, whereas the cells stimulated with LPS, Poly I:C, and R848 showed this reactivity in the nucleus. Interestingly, and supporting the previous results, the nuclear translocation of the p65 subunit mediated by VLPs seems to be time-dependent (Figure 4B), reaching its maximal nuclear translocation at later times.

Taken together, these results indicate that Poly I:C and R848 and, to a lesser extent, VLPs, trigger the NF-κB pathway, which is well correlated with the increased expression of the maturation surface markers on DCs and the release of proinflammatory cytokines.

### 3.4. DCs Maturated by VLPs Containing the RBM from SARS-CoV-2 Spike Protein Induce the Differentiation and Proliferation of T Cells

Autologous MLR was also performed to address whether the DCs maturated by VLPs containing the RBM from SARS-CoV-2 induce T cell differentiation and proliferation. As expected, T cells challenged with mDCs proliferated more than both T cells stimulated with iDCs and T cells without stimuli (T alone). The proliferation was more remarkable when mDCs (Poly I:C and R848) and mDCs (VLPs plus Poly I:C and R848) were used. However, mDCs (VLPs) also induced more proliferation than iDCs (Figure 5A–C), despite the difference between these two conditions not being statistically significant. The proportion between CD4^+^ and CD8^+^ T cells did not change in the presence of any stimuli.

The T cell activation status was analysed through the levels of the cell marker CD25, which is associated with the proliferation and survival of activated T cells [32]. T cells stimulated with all mDCs presented higher expression of CD25 (Figure 5D).

The capability of mDCs to polarize T cells towards Th1, Th2, or Treg phenotypes was assessed by the presence of Tbet^+^ [33], GATA3^+^ [34], and CD25^+^ FoxP3^+^ [35], respectively, within the CD4^+^ T cell population. mDCs induced higher percentages of Th1 cells and a lower polarization towards the Th2 and Treg phenotypes, despite mDCs (VLPs) presenting a tendency for this increment (Figure 5E–G). Accordingly, mDCs (VLPs plus Poly I:C and R848) were the most effective in inducing a Th1-mediated effector immune response.

### 3.5. The Release of IFN-γ by T Cells

Finally, IFN-γ and IL-4 production were investigated in order to reveal the functional capabilities of differentiated T cells. IFN-γ is a cytokine typically produced by activated lymphocytes such as Th1 cells and cytotoxic T cells [36], whereas IL-4 is primarily released by Th2 [37].

IFN- γ release was evident in our data (Figure 6), but IL-4 was not detected in any conditions. T cells stimulated with mDCs (Poly I:C and R848) and mDCs (VLP plus Poly I:C and R848) tend to show a superior capacity to release high amounts of IFN-γ, which strongly correlates with the predominance of Th1 cells. In contrast, VLPs alone do not trigger a high release of IFN-γ, being not statistically different from the condition of T cells + DCs.

## 4. Discussion

DCs have been shown to play a crucial role in several viral infections [38,39,40], including past pandemic coronaviruses [41,42,43]. Moreover, research suggests that evoking T cell-mediated immunity in addition to antibody production will be desired to boost COVID-19 vaccines’ efficacy [44]. In this paper, we elucidated the response of human monocyte-derived DCs (moDCs) to RNA-loaded VLPs containing the RBM from the SARS-CoV-2 S protein. Similar to what was previously described by the team in [17], we performed our experiments using the CuMV VLPs as a whole, since they demonstrated an interesting profile as a vaccine candidate. As controls, we used Poly I:C, a TLR3 agonist, to mimic the interaction and recognition of viral PAMPs, such as the double-stranded ribonucleic acid that originates from the process of viral replication by positive sense-strand DNA and RNA viruses [45]. A TLR7/8 agonist was also used, R848, to simulate the recognition of single-stranded RNA as present in the VLP [46]. We used moDCs, a DC subset that arises from inflammatory conditions [8], because they share key functional similarities with conventional DCs (cDCs) [47] and are easier to obtain [48].

In response to antigen recognition, DCs experience phenotypic changes along with their activation and maturation, which also happens after challenging DCs with the VLPs used. The maturation of DCs triggered by VLPs mimicking viruses, including SARS-CoV-2, has been also reported in previous studies [14,15]. After recognizing the RBM domain via ACE-2 expressed by DCs, the established recognition molecule–ligand complex will be internalized and targeted to lysosomal or late endosomal compartments. Thereafter, the RBM domain is processed for presentation to T cells through MHC molecules [49]. In fact, VLPs downregulated the DCs’ antigen capture activity and promoted DCs’ activation by increasing the expression of co-stimulatory molecules. The up-regulation of MHC II is in concordance with the phenotypic profile of mature DCs. The expression of CD40, CD83, CD86, and MHC II was further boosted with Poly I:C and R848, likely due to the activation of a higher number of PRRs. Thus, RBD from SARS-CoV-2 S protein might be recognized by TLR2, which could dimerize with TLR1 and TLR6 [50] or TLR4 [13]. Therefore, it is straightforward to suppose that VLPs plus Poly I:C and R848 are simultaneously recognized by TLR2, TLR3 and TLR7/8. Once VLPs induce an increase in CxCR4 expression, and this receptor is related to cell migration, it is plausible to speculate that these DCs would be able to migrate towards lymph nodes. Since maturation is a result of antigen recognition, it is reasonable to assume that DCs sensed VLPs. Of note, ACE2 expression levels diminished after DCs encountered VLPs. This may be due to the maturation status of DCs, which is in accordance with a previous study [13]. However, the not-challenged DCs highly expressed ACE2, whereas in a recent study, it was reported that DCs and monocytes have little to no expression of ACE2, suggesting that ACE2 may not be a direct target for SARS-CoV-2 in blood immune cells [51]. Further studies are needed to assess the influence of different differentiation cocktails on the expression of cell receptors.

In accordance with the results achieved for DC activation and maturation, VLPs and Poly I:C and R848 promote higher gene transcription of proinflammatory mediators than VLPs alone. The augmented transcriptional levels of *CCL22* might mediate the crucial crosstalk between T cells and DCs [52]. Accordingly, a previous study reported that the infection of PBMCs with SARS-CoV increases the expression of various CC chemokines and their receptors [53], which corroborates our results due to the similarities observed between SARS-CoV and SARS-CoV-2.

The analysis of cytokines released by DCs shows that both proinflammatory (IL-12p70 and IL-1β) and anti-inflammatory (IL-10) cytokines significantly increase in response to Poly I:C and R848 stimuli. Of great interest is the concordance between the highest transcriptional level and the release of IL-12, which contributes to the efficient polarization of naïve T cells towards the Th1 subtype. Despite the IL-10 gene transcription not being statistically different from the control at the evaluated time point, the cytokine IL-10 is released highly into the medium. This could be explained by the fact that DC-SIGN, upon recognizing the RBD domain, activates a pathway that promotes the acetylation of the p65 NF-kB subunit in the nucleus. This modification increases the activity of the p50-p65 dimer and its retention at the IL-10 promoter. In turn, the activation of this pathway reduces the production of IL-12p70 [49]. Even though we might consider that the higher levels of IL-10 may promote tolerogenic DCs, the molecules expressed on the DCs’ surface refute this hypothesis. Indeed, Islam et al. postulated a possible paradoxical proinflammatory and immunostimulatory role of IL-10 [29]. This hypothesis could explain our results. However, to properly address this question, further studies are needed.

As aforementioned, cellular immunity is a determinant in the control of SARS-CoV-2 infection and underpins COVID-19 vaccine efficacy. We found that the presence of Poly I:C and R848 exacerbated the VLP-induced effector immune responses mediated primarily by IFN-γ-producing Th1. The proliferation of Th1 and CD8^+^ triggered by DCs challenged with VLPs mimicking SARS-CoV-2 was in accordance with a previous study [14]. In fact, effective SARS-CoV-2 control is associated with Th1 cells and a high expression of effector mediators by CD8^+^ T cell [54]. The acute phase of SARS-CoV-2 infection is mainly correlated with Th1 activation, and during the convalescent phase, the antigen-specific T cells can polarize towards a memory phenotype with CD4^+^ and/or CD8^+^ T cells expressing IFN-γ, among other proinflammatory cytokines [55]. Despite its low percentage, our results also suggest a slight presence of Treg in proliferating CD4^+^ T cells. This was as expected due to the high expression of PD-L1 on DCs and the increased levels of *CCL22* gene transcription [56].

Overall, we proved herein that VLPs containing the RBM from the SARS-CoV-2 S protein with two TLR agonists promote the activation and maturation of DCs, and these mature DCs can polarize naïve T cells towards the effector phenotype Th1, with a proliferation of CD4^+^. With this new and deeper insight into the mechanisms triggered by DCs’ activation, this work allows the possibility to conceptualize new designs of effective vaccines against SARS-CoV-2.

## Figures and Tables

**Figure 1 pharmaceutics-15-00825-f001:**
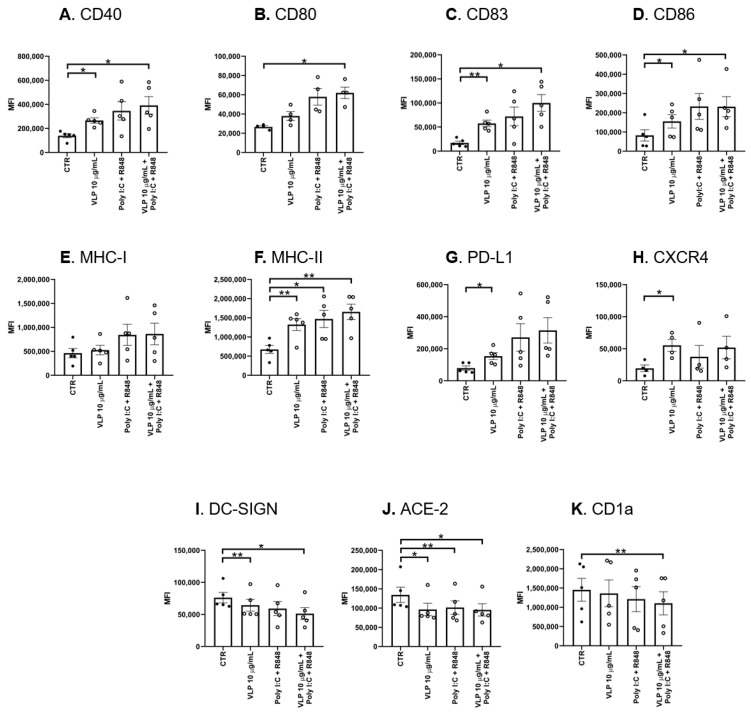
VLPs induce changes in DCs. DCs were treated with 20 μg/mL Poly I:C and 2.5 μg/mL R848, or 10 μg/mL VLPs, or VLPs plus 20 μg/mL Poly I:C and 2.5 μg/mL for 24 h. The expression of (**A**) CD40, (**B**) CD80, (**C**) CD83, (**D**) CD86, (**E**) major histocompatibility complex (MHC) type I (MHC-I), (**F**) MHC type II (MHC-II), (**G**) PD-L1, (**H**) CxCR4, (**I**) DC-DIGN, (**J**) ACE2, and (**K**) CD1a was measured by assessing the mean fluorescence intensity (MFI) by flow cytometry, as described in Materials and Methods. Control cells (CTR) were not stimulated. Each column represents the mean ± SEM of at least four experiments (each point corresponds to an individual donor). * *p* < 0.05; ** *p* < 0.01, relative to the control. VLP: virus-like particles, Poly I:C: polyinosinic:polycytidylic acid, R848: resiquimod.

**Figure 2 pharmaceutics-15-00825-f002:**
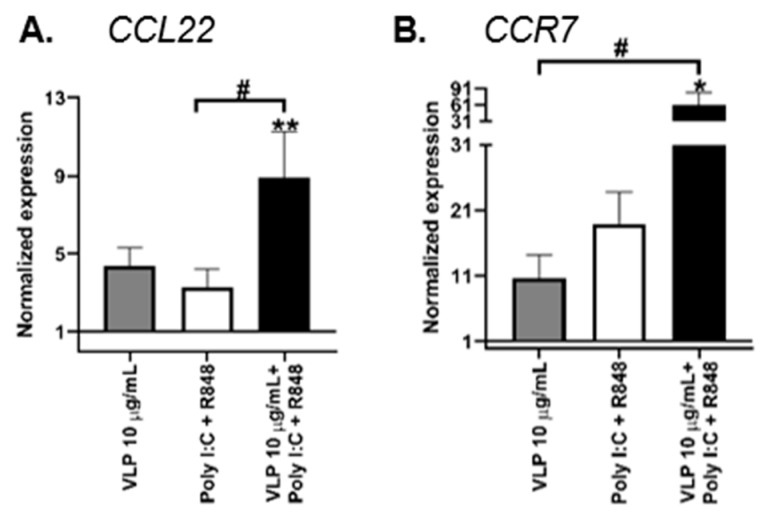
VLPs induce the expression of inflammatory genes in DCs. DCs were treated with 20 μg/mL Poly I:C and 2.5 μg/mL R848, or 10 μg/mL VLPs, or VLPs plus 20 μg/mL Poly I:C and 2.5 μg/mL for 24 h. The expression of (**A**) *CCL22*, (**B**) *CCR7*, (**C**) *IL1β*, (**D**) *IL6*, (**E**) *IL12A*, (**F**) *IL12B*, (**G**) *IL15*, (**H**) *IL18*, (**I**) *iNOS*, (**J**) *TNFα*, (**K**) *IL10*, and (**L**) *TGFβ* genes was assessed using qRT-PCR, as described in Materials and Methods. Each column expresses the normalized expression to the control (not stimulated cells) and represents the mean ± SEM of at least three experiments. * *p* < 0.05; ** *p* < 0.01; *** *p* < 0.001 relative to the control; # *p* < 0.05; ## *p* < 0.01 among the other comparisons. VLP: virus-like particles, Poly I:C: polyinosinic:polycytidylic acid, R848: resiquimod.

**Figure 3 pharmaceutics-15-00825-f003:**
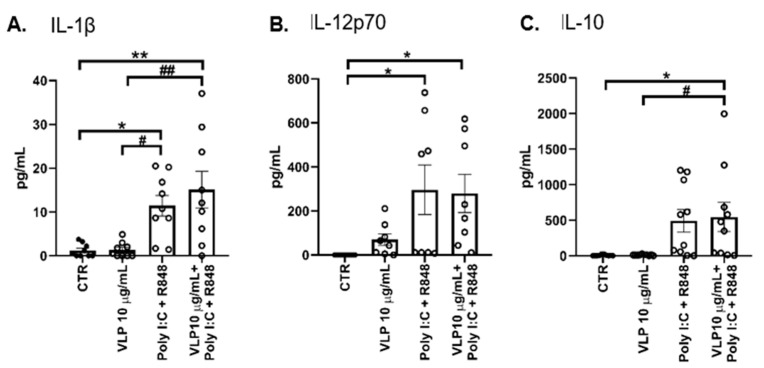
DCs’ release of cytokines into the medium when stimulated with VLPs. DCs were treated with 20 μg/mL Poly I:C and 2.5 μg/mL R848, or 10 μg/mL VLPs, or VLPs plus 20 μg/mL Poly I:C and 2.5 μg/mL for 24 h. The concentration of (**A**) IL-1β, (**B**) IL-12p70, and (**C**) IL-10 was determined through ELISA. Control cells (CTR) were not stimulated. Each column represents the mean ± SEM of at least eight independent experiments (each point corresponds to an individual donor). * *p* < 0.05, ** *p* < 0.01 relative to CTR; # *p* < 0.05, ## *p* < 0.01 among the other comparisons. VLP: virus-like particles, Poly I:C: polyinosinic:polycytidylic acid, R848: resiquimod.

**Figure 4 pharmaceutics-15-00825-f004:**
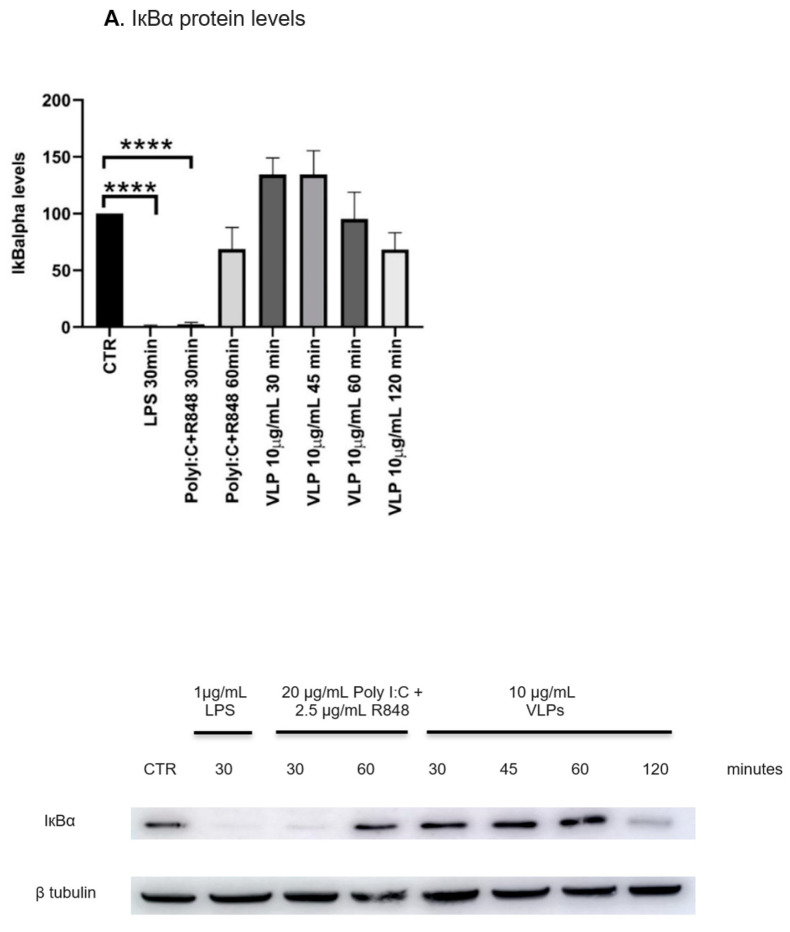
VLPs induce NF-kB nuclear translocation in human DCs. DCs were treated with 20 μg/mL Poly I:C and 2.5 μg/mL R848, or 10 μg/mL VLPs, or VLPs plus 20 μg/mL Poly I:C and 2.5 μg/mL for 24 h. (**A**) IkBα levels were evaluated using Western blot. Each column expresses the normalized expression to the control (not stimulated cells) and represents the mean ± SEM of five independent experiments. Representative images are shown. **** *p* < 0.0001 relative to the CTR. (**B**) Fluorescence staining of the nuclei (blue) and immunofluorescence staining of NF-κB/p65 (green) were performed as described in Materials and Methods. Control cells (CTR) were not stimulated. Scale bar 10 µm. Representative images of each condition are shown. LPS: lipopolysaccharide; VLP: virus-like particles, PolyI:C: polyinosinic:polycytidylic acid, R848: resiquimod.

**Figure 5 pharmaceutics-15-00825-f005:**
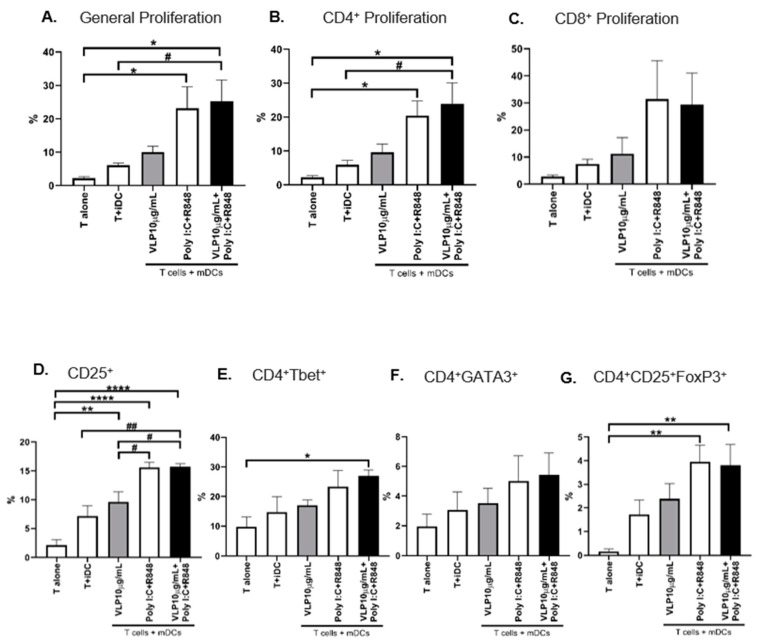
Functional capacities of the stimulated DCs. DCs were stimulated for 24 h with the indicated stimuli and then co-cultured with autologous T cells in a 1:10 ratio. The proliferation of T cells was determined after five days of co-culture by analysing the percentage of the total (**A**) general, (**B**) CD4^+^, and (**C**) CD8^+^ T and cells that presented a decrease in CFSE fluorescence. The percentage of T cell activation was assessed using the expression of the cell marker (**D**) CD25^+^. The polarization of T cells toward (**E**) Th1 (CD4^+^Tbet^+^), (**F**) Th2 (CD4^+^GATA3^+^), and (**G**) “natural” Treg (CD4^+^CD25^+^FoxP3^+^) induced by DCs was evaluated using flow cytometry. The results are expressed as the percentage of cells within T lymphocytes. Each column represents the mean ± SEM of at least three independent experiments. * *p* < 0.05; ** *p* < 0.01, **** *p* < 0.0001 relative to the condition T alone; # *p* < 0.05, ## *p* < 0.01 among the other comparisons. iDC: immature DCs, mDCs: mature DCs, VLP: virus-like particles, PolyI:C: polyinosinic:polycytidylic acid, R848: resiquimod.

**Figure 6 pharmaceutics-15-00825-f006:**
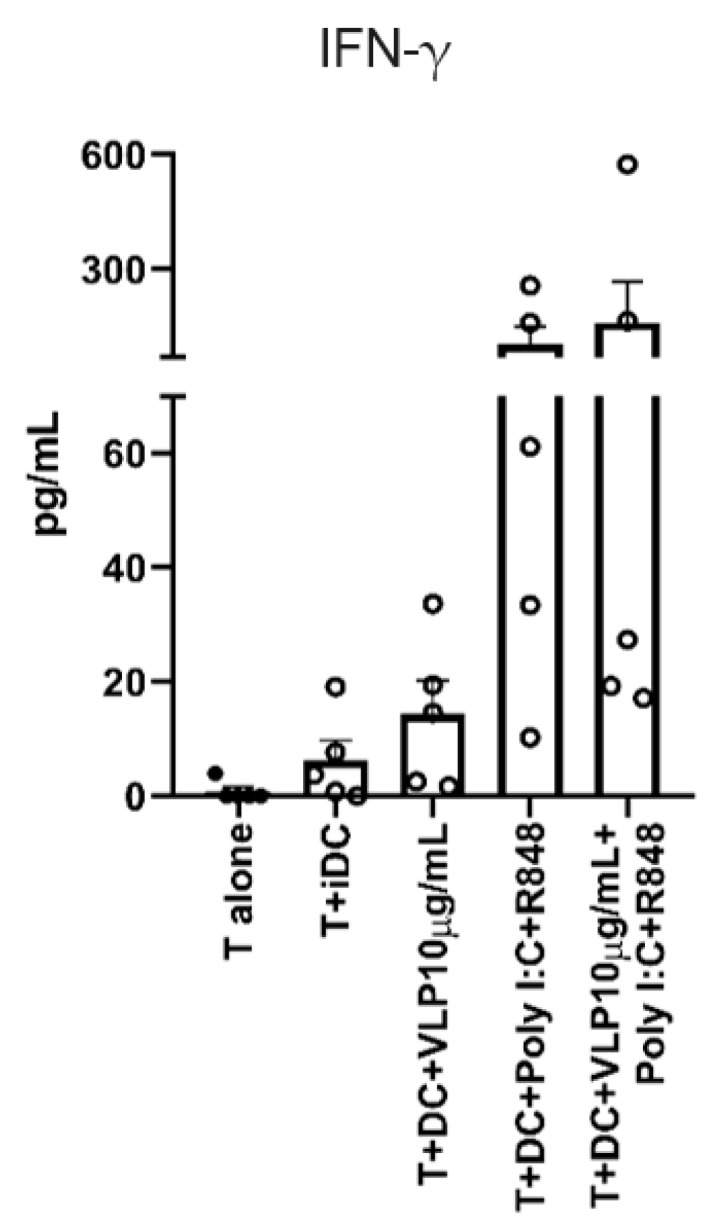
T cells release IFN-γ into the medium. Five days after the co-culture of T cells with stimulated DCs (100,000:10,000), the medium was collected, and the ELISA was performed. Each column represents the mean ± SEM of five independent experiments (each point corresponds to an individual donor). No statistically significant differences were found in any parameter analysed. iDC: immature DCs, mDCs: mature DCs, VLP: virus-like particles, PolyI:C: polyinosinic:polycytidylic acid, R848: resiquimod.

## Data Availability

The datasets used and/or analysed during the current study are available from the corresponding author on reasonable request.

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
