# Peer review of "CuMV VLPs Containing the RBM from SARS-CoV-2 Spike Protein Drive Dendritic Cell Activation and Th1 Polarization"

_pharmaceutics, 2023, doi:10.3390/pharmaceutics15030825_

Round 1
Reviewer 1 Report
The paper is interesting, but there are some concerns that need to be corrected. My comments are as follows.
(1) The document on Line 190 is incomplete.
(2) It is unclear to call something that expresses RBM on a cucumber mosaic virus host as a SARS-CoV-2 VLP (virus-like particle). It is uncertain whether the same results can be achieved by a VLP created from animal cells expressing S, M, N, E, which is commonly done. If investigating the immune response of a cucumber mosaic virus host as a vaccine candidate, it would be better to change the title.
(3) The method of obtaining the negative control in the experiment is incorrect. The experimental data of the negative control VLP without expression of SARS-CoV-2 RBM should be used as a control. In every figures, you need to show whether cellular responses are dependent on RBM.
Author Response
"Please see the attachment."

Reviewer 2 Report
The authors investigated the role of virus like particles (VLPs) produced in monocyte derived dendritic cells (DCs) in triggering the immunity to SARS-CoV-2 virus. The statistical analysis is correct and the data are mainly well presented and discussed. My only criticism regards the discussion of the non statistical significant data in the discussion section. Overall the manuscript quality is very good.
Reviewer 3 Report
This is a very detailed paper that describes studies on the impact of virus like particles that are expressing? the receptor binding motif from the SARS-CoV-2 S protein on expression of a number of immunoregulatory molecules in human dendritic cells.
The introduction to this paper was short but did give context to the work. I would have liked this to include some of the information reported in previous studies here - refs 46&47. These are included in the discussion but would be good background. I was pleased to see an excellent list of abbreviations - this was vital for someone like me who is not fully embedded in immunological nomenclature. I feel that the conclusions of the work were well founded, and I thought the flow of experiments and their rational was clearly described. However, I was little concerned about the use of the word tendency line 503 and also lines 333/334 increased levels.... despite not displaying statistical significance - see issues below.
Issues to note.
Methods
Lines 189-190 this is incomplete and presumably describes in detail the 'phenotype' of the VLP's.
Not sure I could find in the methods why they were using R848?
Results
The authors need to review phrases around statistical significance - lines 333/334 and 386-388 - this also impacts on the discussion.
Minor points
Lines 188 & 327 - need Escherichia coli in full first time then E.coli - must all be italicised
Line 315 - should be Tukey....
Figure 2 legend -line 394 induce in what?
Figure 3 legend -release of ...
Round 2
Reviewer 1 Report
The revised manuscript has adequately addressed the comments by reviewers. It has been improved and is now acceptable.